# A Whole-Genome Assembly for *Hyaloperonospora parasitica*, A Pathogen Causing Downy Mildew in Cabbage (*Brassica oleracea* var. *capitata* L.)

**DOI:** 10.3390/jof9080819

**Published:** 2023-08-03

**Authors:** Yuankang Wu, Bin Zhang, Shaobo Liu, Zhiwei Zhao, Wenjing Ren, Li Chen, Limei Yang, Mu Zhuang, Honghao Lv, Yong Wang, Jialei Ji, Fengqing Han, Yangyong Zhang

**Affiliations:** 1State Key Laboratory of Vegetable Biobreeding, Institute of Vegetables and Flowers, Chinese Academy of Agricultural Sciences, Beijing 100081, China; wuyk1129@163.com (Y.W.); 13126720352@163.com (B.Z.); 17863805323@163.com (W.R.); 18205480752@163.com (L.C.); yanglimei@caas.cn (L.Y.); zhuangmu@caas.cn (M.Z.); lvhonghao@caas.cn (H.L.); wangyong@caas.cn (Y.W.); jijialei@caas.cn (J.J.); 2China Vegetable Biotechnology (Shouguang) Co., Ltd., Shouguang 262700, China; liushaobosdau@163.com (S.L.); zgys9814@126.com (Z.Z.)

**Keywords:** *Hyaloperonospora parasitica*, genome sequence resource, downy mildew, cabbage

## Abstract

*Hyaloperonospora parasitica* is a global pathogen that can cause leaf necrosis and seedling death, severely threatening the quality and yield of cabbage. However, the genome sequence and infection mechanisms of *H. parasitica* are still unclear. Here, we present the first whole-genome sequence of *H. parasitica* isolate BJ2020, which causes downy mildew in cabbage. The genome contains 4631 contigs and 9991 protein-coding genes, with a size of 37.10 Mb. The function of 6128 genes has been annotated. We annotated the genome of *H. parasitica* strain BJ2020 using databases, identifying 2249 PHI-associated genes, 1538 membrane transport proteins, and 126 CAZy-related genes. Comparative analyses between *H. parasitica, H.arabidopsidis*, and *H. brassicae* revealed dramatic differences among these three Brassicaceae downy mildew pathogenic fungi. Comprehensive genome-wide clustering analysis of 20 downy mildew-causing pathogens, which infect diverse crops, elucidates the closest phylogenetic affinity between *H. parasitica* and *H. brassicae*, the causative agent of downy mildew in *Brassica napus*. These findings provide important insights into the pathogenic mechanisms and a robust foundation for further investigations into the pathogenesis of *H. parasitica* BJ2020.

## 1. Introduction

Downy mildew is an important disease that seriously affects the economics of horticultural crop production, such as *Spinacia oleracea*, *Brassica oleracea*, *Brassica rapa*, *Brassica napus*, *Cucumis sativus*, and *Vitis vinifera* [1,2,3,4]. Cabbage (*Brassica oleracea* var. *capitata* L.) is an important vegetable cultivated in many countries around the world due to its high economic and nutritional values. However, cabbage can be infected by a variety of pathogens, which can lead to a series of diseases including clubroot, fusarium wilt, black rot, etc. [2,5,6]. In addition, downy mildew, caused by the oomycete *Hyaloperonospora parasitica*, has become a serious threat to cabbage production in recent years [7]. *Hyaloperonospora parasitica* spreads in the field through conidiospores. During cabbage production, downy mildew usually occurs in spring and autumn. Cold temperatures and high humidity environment provide favorable conditions for outbreaks of cabbage downy mildew. Under suitable conditions, the conidia of downy mildew can spread quickly in a field with the circulation of rain and air [8]. Downy mildew can damage cabbage from the cotyledon stage to the adult stage and can infect the stems, rosette leaves, head leaves, and seed pods [9]. Infection of downy mildew can cause the death of over 75% of seedlings and more than 50% yield losses [2]. Protected cultivation of cabbage has increased in recent years, because it can supply cabbage much earlier than open field cultivation, especially in the early spring. However, indoor production also provides an appropriate environment for *Hyaloperonospora parasitica* conidia germination [7]. As obligate biotrophs, the pathogens causing downy mildew have gone through several classification changes. Most recently, Constantinescu summarized that *H. parasitica* belongs to the phylum *Oomycota*, order *Peronosporales*, family *Peronosporaceae*, and genus *Hyaloperonospora* [10]. This genus includes *Hyaloperonospora arabidopsidis*, which causes *Arabidopsis* downy mildew, as well as *Hyaloperonospora brassicae*, which causes *Brassica napus* downy mildew, etc.

In recent years, whole-genome sequencing has become common due to the decrease of cost and improvement of sequencing technologies [11]. The de novo sequencing of fungi and bacteria genome can produce complete draft sequences, which facilitate researchers to mine key pathogenic agents of these pathogens, understand their pathogenesis, and provide insights to develop disease-resistant varieties. Previous studies have shown that genome assembly of pathogenic fungi is significant for exploring the infection mechanism, as demonstrated in studies on *Magnaporthe oryzae* [12], *Stagonospora tainanensis* [13], and *Setosphaeria turcica* [14]. However, there have been few genomic studies conducted on the causative pathogen of Brassicaceae downy mildew. Until now, only three genome sequences have been reported in this group: one for *Hyaloperonospora arabidopsidis*, causing *Arabidopais thaliana* downy mildew, and two for *Hyaloperonospora brassicae*, causing *Brassica napus* downy mildew [15,16].

Here, we report the first draft assembly of *H. parasitica,* which causes downy mildew in cabbage, providing a resource for analyzing the pathogenic factors and infection mechanisms of *H. parasitica*.

## 2. Materials and Methods

### 2.1. Purification of H. parasitica

*H. parasitica* strain BJ2020 was isolated from the cabbage inbred line “2020-w5”, cultivated in the greenhouse of the Institute of Vegetables and Flowers, Chinese Academy of Agricultural Sciences, Beijing, China. Fresh downy mildew conidia were isolated from naturally infected leaves in the field using a method as previously described [17]. Subsequently, a conidial suspension was sprayed on the seedlings of “2020-w5”. After being grown in a greenhouse under a 16 h light/8 h dark cycle for 6 d and then placed under high humidity in the dark for 24 h, the newly formed sporangia will germinate [18]. The inoculated plants showed heavy necrosis with sporulation dispersed over the entire leaf surface [19]. The above procedure was repeated several times, and finally, the purified *H. parasitica* isolate BJ2020 was obtained.

### 2.2. Library Construction and Sequencing

A NucleoBond^®^ HMW DNA kit (MN NucleoBond, Düren, Germany, 740160.20) was used for high-quality genome extraction from samples. DNA concentrations and purity were determined with a Qubit 4.0 instrument (Thermo, Q33226). DNA integrity was assessed by 0.75% agarose gel electrophoresis. The whole-genome DNA was randomly fragmented to an average size of 200–400 bp. The selected fragments were subjected to end repair, 3′ adenylation, adapter ligation, and PCR amplification. After purification with magnetic beads, the library was qualified with a Qubit 4 fluorometer (Thermo, Waltham, MA, USA, Q33226), and the length of the library inserts was assessed by 2% agarose gel electrophoresis. The qualified libraries were sequenced on the Illumina NovaSeq 6000 platform with about 300 bp reads at Sangon Biotech (Shanghai, China).

### 2.3. De Novo Genome Assembly

After sequencing, raw reads were filtered with Trimmomatic (v0.36) by removing adaptors and low-quality reads, and clean reads were obtained [20]. Genome assembly was performed using SPAdes (v3.15), and GapFiller (v1.11) was used to fill gaps to generate a genome assembly of strain BJ2020 [21,22]. SPAdes (default parameters) was used for sequence error correction, assembly using multiple Kmer values based on read lengths, and synthesis of the assembly results for each Kmer value to obtain the best result. Then, GapFiller was used for GAP supplementation in the spliced contigs, and PrInSeS-G was finally used for the sequence correction of editing errors and the insertion and deletion of small fragments in the splicing process [23]. The benchmark universal single-copy orthologs (BUSCO), version 5.2.2, were employed to assess the genome assembly completeness with respect to fungal ancestry [24].

### 2.4. Gene Prediction and Functional Annotation

For de novo gene prediction, annotations were generated using GeneMark-ES, an algorithm utilizing models parameterized by unsupervised training, with the fungi mode [25]. Thereafter, BLAST searches were conducted for all protein-coding genes in the National Center for Biotechnology Information (NCBI, https://www.ncbi.nlm.nih.gov/, 7 June 2021) databases. The whole genome including repeat elements was annotated, including the Conserved Domain Database (CDD), euKaryotic Ortholog Groups (KOG), Clusters of Orthologous Groups of proteins (COG), NCBI nonredundant protein sequence (NR), NCBI nucleotide sequence (NT), SwissProt (a manually annotated and reviewed protein sequence database), TrEMBL and other databases.

The Pathogen Host Interactions Database (PHI-base) has been utilized as a valuable resource for predicting key genes involved in the interaction between pathogens and their hosts [26]. The Carbohydrate-Active enZYmes Database (CAZy) was used to annotate carbohydrate active enzyme-encoding genes [27]. The Transporter Classification Database (TCDB) was used to predict membrane transport proteins [28].

### 2.5. Identification of RNAs and Repeated Sequences

tRNAscan-SE was used to annotate transfer RNAs (tRNAs) with eukaryotic parameters [29]. The ribosomal RNAs (rRNAs) were annotated using RNAmmer [30], and the small nuclear RNAs (snRNAs) were predicted by comparison with Rfam [31].

RepeatModeler was used for the de novo prediction of repeated sequences among the assembly results, and RepeatMasker (http://repeatmasker.org/, 7 June 2021) was used to identify where and how often each type of repeat occurred in a segment of the genome [32].

### 2.6. Secretome and Effector Identification

We used the SignaIP v5.0 Server to identify the proteins with signal peptides, after which the TMHMM Server v1.0.10 was employed to remove the proteins with transmembrane domains [33,34]. The remaining proteins were considered putative secreted proteins. Then, EffectorP-3.0 was used to annotate the effectors of the secretome [35].

### 2.7. Genomic Comparison and Phylogenomic Analysis

The genome data of *H. arabidopsidis* were obtained from Ensembl (https://protists.ensembl.org/, 8 June 2021). The data on *H. arabidopsidis*, *H. brassicae* and the whole-genome sequences of other fungal species were downloaded from NCBI. Then, Genemark was used to predict the CDS regions, and TBTOOLS was employed to translate the CDS into proteins [36]. Orthofinder v2.5.4 was used to construct a phylogenomic tree of species with the help of MAFFT and FASTTREE [37,38,39]. Then, single-copy genes from ten species were used to construct a phylogenomic tree based on evolutionary time. All phylogenomic trees were constructed with Interactive Tree Of Life (iTOL) v6.5.8 online services [40].

## 3. Results

### 3.1. Genome Assembly of Strain BJ2020

The genome of BJ2020 was sequenced on the Illumina MiSeq platform. After sequencing, raw reads were filtered via Trimmomatic (v0.36) by removing adaptors and low-quality reads, and a total of 6.67 Gb of clean reads was obtained, which was equivalent to a 179-fold sequencing depth [20]. SPAdes (v3.15) was used to de novo assemble the clean reads, and GapFiller (v1.11) was used to fill gaps in raw assembly sequences. Then, we used PrinSeS-G to complete sequence correction [21,22,23]. Finally, we obtained a genome size of 37.10 Mb with an N50 of 20,542 bp and a CG percentage of 51% (Table 1). The assembled genome sequences were processed for further analysis and functional annotation. The assessment of genome completeness indicated that out of a total of 255 orthologous BUSCO genes, 235 (92.1%) were identified as complete and single-copy orthologs. Additionally, eight (3.1%) duplicated genes and seven (2.7%) fragmented genes have also been identified.

We used Genemark, tRNAscan-SE, and RepeatModeler, respectively, to predict protein-coding genes, RNAs, and repeat sequences. Results showed that the genome contained 9991 protein-coding genes, 237 transfer RNAs, and 13 ribosomal RNAs, with 11,653,830 bp repeat regions.

Gene repeat analysis showed that the repeat sequences of *H. parasitica* strain BJ2020 account for 31.41% of the genome. These repeat sequences consisted of DNA repeats (0.68%), long interspersed nuclear elements (LINEs, 1.98%), long terminal repeats (LTRs, 20.66%), low complexity repeats (0.07%), simple repeats (1.14%), and some unknown sequences (6.88%). In common with the plant genome, LTR is the most common type of repeat sequence in the genome of *H. parasitica* strain BJ2020.

### 3.2. Gene Functional Annotation

The functional annotation of the 9991 predicted genes was conducted using the NCBI nonredundant protein database (5280 genes, 52.85%), Conserved Domain Database (5447 genes, 54.52%), euKaryotic Ortholog Group database (3915 genes, 39.19%), Protein family database (4829 genes, 48.33%), Swiss-Prot (5181 genes, 51.86%), TrEMBL (5223 genes, 52.28%), Gene Ontology database (5051 genes, 50.56%), and Kyoto Encyclopedia of Genes and Genomes (2254 genes, 22.56%). A total of 6128 genes were annotated in at least one database, accounting for 61.34% of the total predicted genes (Table 2).

GO and KEGG analysis for the strain BJ2020 genome revealed that the top five enriched GO terms were cellular process, cell, cell part metabolic, and process and binding, with 3837 genes, 3535 genes, 3534 genes, 3500 genes, and 3263 genes, respectively. Most identified genes were involved in biological processes (Figure 1). In all 31 KEGG pathways, the top five pathways were ‘signal transduction’, ‘cell growth and death’, ‘endocrine system’, ‘translation’, and ‘carbohydrate metabolism’, including 547 genes, 452 genes, 442 genes, 377 genes, and 351 genes, respectively (Table 2, Figure 2). The enrichment analysis of the top five KEGG pathways suggests that the majority of genes in strain BJ2020 are likely involved in crucial cellular processes, including signal transduction, cell growth and death, and endocrine signaling, as well as fundamental metabolic pathways such as translation and carbohydrate metabolism.

There were 3915 genes annotated in the KOG database, among which the most genes were annotated to ‘General function prediction only’ (533), accounting for 13.6% of the total number of KOG annotations, followed by 414 genes annotated to ‘posttranslational modification, protein turnover, chaperones’, 355 genes annotated to ‘signal transduction mechanisms’, 319 genes annotated to ‘Function unknown’, and 285 genes annotated to ‘Translation, ribosomal structure and biogenesis’ (Figure 3). The abundance of genes annotated to ‘General function prediction only’ suggests that many genes in the strain BJ2020 may have not been fully characterized or are involved in basic cellular functions. Meanwhile, the significant number of genes annotated to ‘posttranslational modification, protein turnover, chaperones’, ‘signal transduction mechanisms’, and ‘Translation, ribosomal structure and biogenesis’ indicate that these processes play important roles in the biology of the strain BJ2020. The large number of genes with ‘Function unknown’ annotation highlights the need for further investigation and characterization of these genes to fully understand their biological significance.

### 3.3. Identification of Disease-Related Genes

Among the top three Pfam annotations, two originated from abundant repeat elements: ‘Reverse transcriptase2’ and ‘Reverse transcriptase1’, with 206 and 130 genes, respectively. Additionally, other repeat-encoded genes, such as ‘Integrase core domain’ (92 genes) and ‘gag-polypeptide of LTR copia-type’ (77 genes), were also detected (Figure 4A).

We identified a series of disease course-related genes, including 126 CAZys, 688 signal peptides, 1538 membrane transporters, and 2249 pathogenicity-related proteins (PHIs) (Figure 4, Appendix A). The identified CAZys included 58.46% glycoside hydrolases (GHs), 44.35% glycosyl transferases (GTs), 9.70% auxiliary activities (AAs), 6.50% carbohydrate-binding modules (CBMs), 5.40% polysaccharide lyases (PLs), and 4.30% carbohydrate esterases (CEs). These proportions of different CAZys suggest that the identified enzymes may have diverse roles in the degradation and modification of carbohydrates, including the breakdown of complex plant cell wall materials and the modification of glycans on proteins and lipids. The comprehensive analysis of these CAZys could provide important insights into their functions and potential applications in various biotechnological and industrial processes. A total of 1538 membrane transport proteins of BJ2020 also have been identified. The top five membrane transport proteins were ‘The 5’-AMP-activated protein kinase (AMPK) Family’ (224), ‘The NEAT-domain containing methaemoglobin heme sequestration (N-MHS) Family’ (96), ‘The Calmodulin Calcium Binding Protein (Calmodulin) Family’ (92), ‘The Ezrin/Radixin/Moesin-binding Phosphoprotein’ (79), ‘The Outer Membrane Factor (OMF) Family’ (63), and ‘(EBP50) Family’ (50) (Appendix A). This may imply that the onset of downy mildew infestation is closely related to the AMPK pathway, and it may disrupt the host’s PAMP-triggered immunity system by affecting the stability of the calcium contents of host cells.

We identified 688 proteins with signal peptides and 663 proteins with extracellular locations that were considered putative secreted proteins after the removal of proteins containing transmembrane helixes. Among these putative secreted proteins, we identified 224 cytoplasmic effectors and 52 apoplastic effectors. The number of cytoplasmic effectors far exceeded the number of apoplastic effectors (Figure 4D). Further analysis of these cytoplasmic effectors can help us better understand their functions and their roles in the interaction between the pathogen and host.

Analysis of the intersection between *H. parasitica* strain BJ2020 secreted protein genes, PHI genes, and CAZy gene annotations revealed 35 shared genes (Appendix A). Our analysis of *H. parasitica* strain BJ2020 secreted protein characteristics and two database annotations suggests that the overlapping genes may play a critical role in cabbage infection by *H. parasitica*.

Additionally, based on the analysis of secreted proteins, we identified 65 effectors containing the RxLR motif (Appendix A). Phylogenomic analysis of these RxLR effectors classified them into three clusters (Appendix A).

### 3.4. Comparison between H. parasitica and Other Brassicaceae Crop Downy Mildew Pathogens

Downy mildew is an important disease of Brassicaceae crops, but few genomic studies of the pathogens have been reported. Among Brassicaceae crops, only the genomic resources of *H. arabidopsidis*, which causes downy mildew in *Arabidopsis*, as well as *H. brassicae*, which causes downy mildew in *Brassica napus*, have been reported [16,41]. By referencing the genome assembly methods used for *H. brassicae* and *H. arabidopsidis*, we performed multiple rounds of de novo assembly for *H. parasitica*, significantly improving the quality of the assembly. In the assembled *H. parasitica* strain BJ2020 genome, a total of 4631 contigs were obtained, with the longest contig having a length of 156,777 bp. Here, the genome sequences were compared among *H. parasitica*, *H. brassicae*, and *H. arabidopsidis* (Table 3). Although all these three pathogens cause Brassicaceae downy mildew, there are considerable differences in their genomes. *H. parasitica* had the smallest genome but showed the highest GC content in the genome. *H. brassicae* Sample B and Sample C were two *Brassica napus* downy mildew pathogens with differences in virulence, and their genomes contained the largest number of genes encoding proteins. Additionally, the genome of Sample C was the largest reported genome among Brassicaceae crop downy mildew pathogens. *H. arabidopsidis* had the longest N50 and the most contigs, but the lowest number of protein-coding genes.

### 3.5. Phylogenomic Analysis

To elucidate the evolutionary relationships among various downy mildew pathogens affecting different crops such as *Arabidopsis*, grapevine, cucumber, and others, we conducted a comprehensive genome-wide clustering analysis of homologous genes for 20 downy mildew-causing pathogens. These pathogens vary in terms of host specificity and genome size, but all of them inflict significant damage to the productivity of their hosts (Appendix A). The analysis demonstrated precise clustering of pathogens from diverse genus into distinct branches (Figure 5). The *Peronospora effusa*, causing downy mildew in spinach, and the *Peronospora tabacina*, causing downy mildew in tobacco, clustered separately and distinctly from the other 18 downy mildew-causing pathogens. Additionally, phytopathogenic races of different pathogens infecting the same host were found to cluster together within the same branch, indicating their close evolutionary association. Interestingly, in comparison to the *H. parasitica* strain BJ2020, which infects cabbage, the two *H. brassicae* strains infecting *B. napus* and the three *H. arabidopsidis* strains infecting *Arabidopsis* exhibit relatively smaller variations in genome size. However, the phylogenetic analysis indicates a closer evolutionary relationship between *H. brassicae* and *H. parasitica*, as opposed to *H. arabidopsidis*. On the other hand, the clustering of three different hosts’ downy mildew pathogens, *H. parasitica*, *H. brassicae*, and *H. arabidopsidis*, belonging to the genus *Hyaloperonospora*, onto three distinct branches suggests varying infectivity capabilities of these three pathogens towards different hosts within the Brassicaceae family, implying their coevolutionary dynamics with their respective hosts.

## 4. Discussion

The occurrence of downy mildew is species-specific, and the pathogens of downy mildew differ among different host plants. As sequencing costs have reduced, genome sequencing has become an important tool for studying fungal pathogenicity, heredity, and evolution and has facilitated the identification of the different pathogenic species responsible for downy mildew on different host plants. The genome of the *H. arabidopsidis* strain Emoy2 had been successfully assembled, and the results indicated that the numbers of RxLR effectors in obligate biotrophs were evolutionarily significantly reduced compared to those in other fungi [41]. In this study, we assembled the first draft genome of *H. parasitica* strain BJ2020, which causes cabbage *(Brassica oleracea* var. *capitata* L.) downy mildew.

The genome size of *H. parasitica* strain BJ2020 is an important factor in understanding its pathogenicity and genetic makeup. With a genome size of 37,102,749 bp, this strain contains a significant number of protein-coding genes (9991), tRNAs (237), rRNAs (13), and repeat sequences (31.41%). Through gene function annotation analysis, 5051 genes from *H. parasitica* strain BJ2020 were assigned to GO categories, and 2254 genes were assigned to KEGG categories, providing a better understanding of the genetic functions of the organism. These findings provide valuable information for further research on *H. parasitica* critical pathogenic effectors and pathogenesis. Interestingly, there were significant differences in the genome of *H. parasitica* strain BJ2020 compared to other strains such as *H. arabidopsidis* strain Emoy2 and *H. brassicae* strains Sample B and Sample C [16,41]. These differences may provide insights into the evolution of *H. parasitica* and its adaptation to different host plants. The genome analysis of *H. parasitica* strain BJ2020 provides a foundation for future studies on this important plant pathogen. The information gathered through gene function annotation and genome comparison may lead to the identification of new pathogenicity factors and the development of novel strategies for disease management.

The significant differences in genome size and gene number among pathogens *H. parasitica*, *H. brassicae*, and *H. arabidopsidis* may imply significant differences in their life history and adaptation to the environment. They may rely on different genes and pathways to adapt to their environment and hosts, and the difference in gene number may reflect differences in host affinity, life cycle, metabolic pathways, etc. Also, the sequenced and assembled genome is at the scaffold level, which may have missed small gaps or dispersed pseudogenes as reported [42]. For *H. parasitica* strain BJ2020, its genome has the lowest N50 proportion, the fewest contigs, and the smallest genome size among the three pathogens, despite causing the same disease in Brassicaceae crops. These features may indicate that *H. parasitica* strain BJ2020 has undergone genome reduction and has adapted to host and environmental pressures. This also suggests that *H. parasitica* strain BJ2020 may have reduced redundant genes, minimized intergenic regions, and deleted unnecessary genome components such as junk DNA during the adaptation process. These findings provide a theoretical basis for a better understanding of the genetic mechanisms and ecological adaptability of these pathogens and can aid in the development of effective disease control strategies.

The cell wall is the first line of defense in plants against pathogenic pathogen infestation. The CAZys of pathogenic pathogens coevolve with the plant cell wall during the process of fighting against pathogens in plants [43,44]. Here, 126 CAZys were predicted, which were assigned to six types (GHs, GTs, AAs, CBMs, PLs, and CEs). GH, PL, and CE CAZys are major member involved in the degradation of plant cell walls [45]. The strain BJ2020 was rich in GH and GT CAZys, which may participate in the puncture of plant cell wall. It is reported that GH and GT class gene were mainly related to the degradation and synthesis of chitinase, cellulase, and hemicellulase to affect plant cell walls [46].

The effectors play a critical role in the infection of plants by pathogenic pathogens. In recent years, many pathogenic effectors of pathogenic pathogens have been identified [41,47]. Pathogenic effectors can help recognize and colonize the host plants [48]. Here, we obtain 663 putative secreted proteins, which include 224 cytoplasmic effectors and 52 apoplastic effectors. Apoplastic effectors may function as enzyme inhibitors, apart from helping pathogens escape identification of plant’s immune system and scavenge molecules that trigger immune responses [49]. Cytoplasmic effectors may act as a target or transfer of pathogenic pathogens [50,51]. Furthermore, 2249 PHIs had been identified, which were important for pathogenicity of *H.parasitica* strain BJ2020. All these results can help build a bridge for probing into the infection mechanism of *H. parasitica* strain BJ2020.

## 5. Conclusions

In this study, we obtained a high-quality reference genome of *H. parasitica*. The whole-genome sequence of *H. parasitica* BJ2020 provides an important reference for subsequent transcriptomic, proteomic, and metabolomic research. Moreover, we identified 9991 protein-coding genes, 237 tRNAs, and 13 rRNAs. GO, KEGG, and KOG annotation among these protein-coding genes indicated that most of the genes in the *H. parasitica* BJ2020 genome are related to cellular processes. Annotation results from the PHI database, CAZy database, and other feature databases implied that reverse transcription genes may play important roles in the interaction between *H. parasitica* and host plants. Our analysis of *H. parasitica* strain BJ2020 secreted protein and PHI database and CAZy database annotations suggested that the overlapping 35 genes and 65 RxLR effectors may represent critical factors for cabbage infected by *H.parasitica*. This research enriched the resources of cabbage downy mildew and provided a theoretical basis for the subsequent study of *H. parasitica* infection and the mechanism of cabbage downy mildew disease resistance.

## Figures and Tables

**Figure 1 jof-09-00819-f001:**
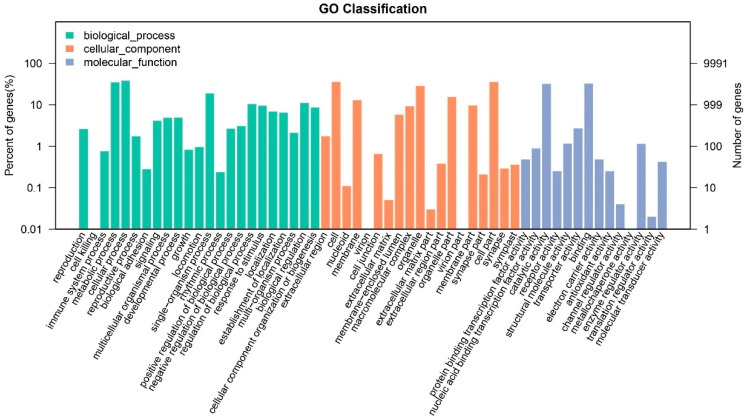
GO annotation of the genome of *H. parasitica* strain BJ2020.

**Figure 2 jof-09-00819-f002:**
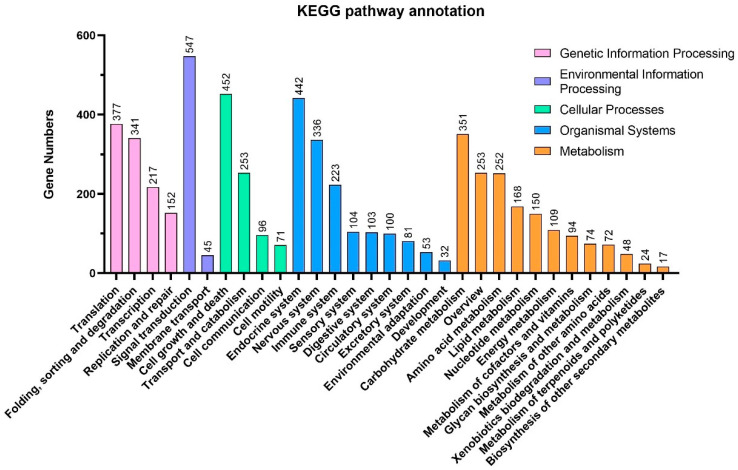
KEGG pathway annotation of the genome of *H. parasitica* strain BJ2020.

**Figure 3 jof-09-00819-f003:**
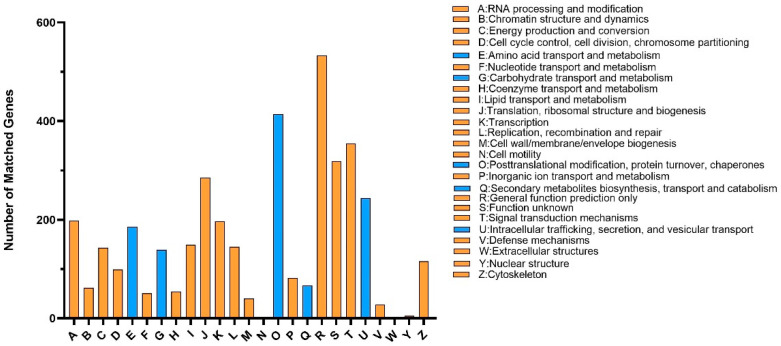
KOG annotation of the genome of *H. parasitica* strain BJ2020.

**Figure 4 jof-09-00819-f004:**
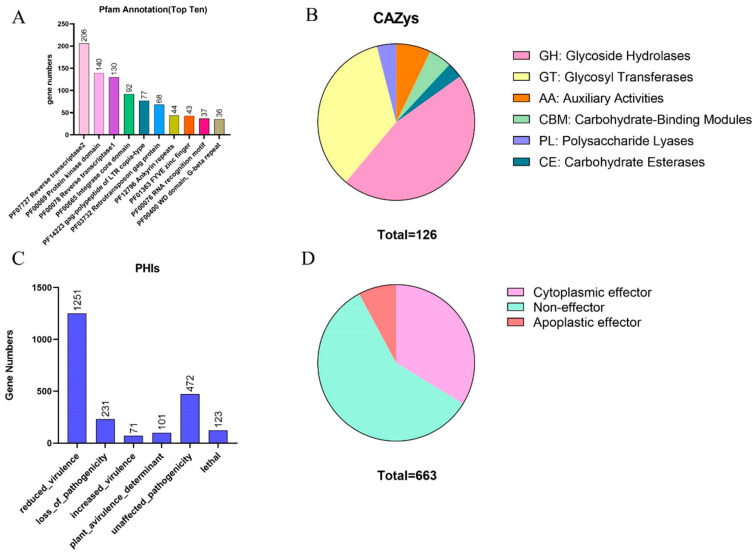
Summary of pathogenicity-related gene annotations. (**A**) PFAMs, (**B**) CAZys, (**C**) PHIs, and (**D**) putative secreted proteins.

**Figure 5 jof-09-00819-f005:**
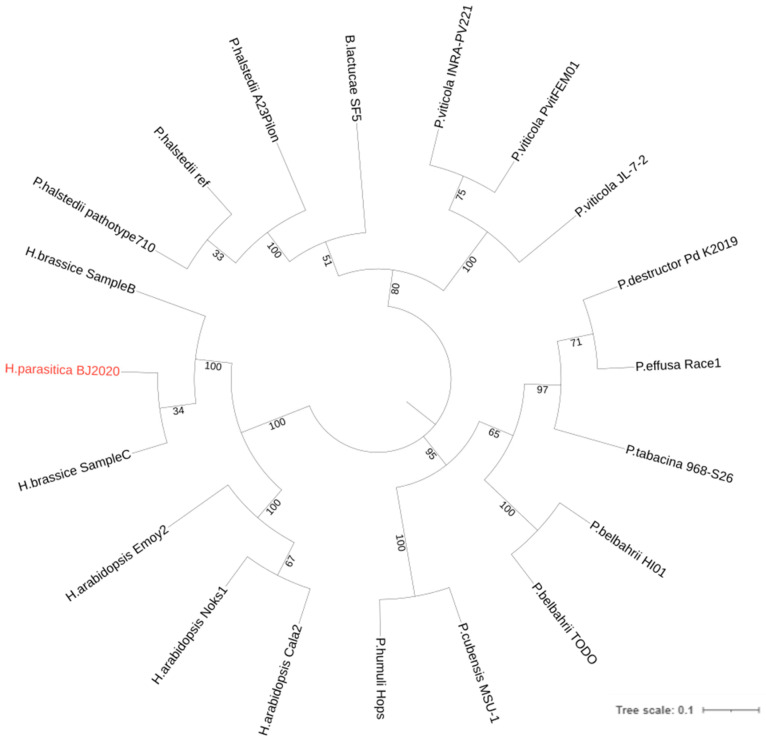
Comparative genomic analysis with 20 downy mildew-causing pathogens.

**Table 1 jof-09-00819-t001:** Genome statistics of *H. parasitica* isolate BJ2020.

Features	BJ2020
Genome size (bp)	37,102,749
Number of contigs	4631
N50 (bp)	20,542
GC content (%)	51%
Protein-coding genes	9991
Gene density (number of genes per Mb)	269
Min length (bp)	118
Max length (bp)	18,240
Average length (bp)	1191.66
Total coding gene length (bp)	11,905,868
tRNA	237
rRNA	13
Repeat regions (bases)	11,653,830
Repeat ratio (%)	31.41%
Simple repeats	8712

**Table 2 jof-09-00819-t002:** Statistical results of gene functional annotation in functional databases of *H. parasitica* BJ2020.

Database	Number of Genes	Percentage
CDD	5447	54.52%
KOG	3915	39.19%
NR	5280	52.85%
PFAM	4829	48.33%
SwissProt	5181	51.86%
TrEMBL	5223	52.28%
GO	5051	50.56%
KEGG	2254	22.56%
Annotated in at least one database	6128	61.34%
Annotated in all databases	1955	19.57%
Total Unigenes	9991	100.00%

**Table 3 jof-09-00819-t003:** Statistical results of comparison between the genome of *H. parasitica*, *H. brassicae*, and *H. arabidopsidis*.

	*H. parasitica*	*H. brassicae*		*H. arabidopsidis*
Strain	BJ2020	Sample B	Sample C	Emoy2
Total size	37.10 Mb	79.39 Mb	92.19 Mb	78.38 Mb
Protein coding genes (≥250 bp)	9991	36,819	40,346	14,321
Number of contigs	4631	6438	6470	10,486
N50	20.5 Kb	23.5 Kb	24.5 Kb	41.9 Kb
GC (%)	51%	47%	47%	47%

## Data Availability

The whole-genome sequence of *Hyaloperonospora parasitica* isolate BJ2020 has been deposited in NCBI GenBank with BioProject number PRJNA907175 under accession number JAPPWB000000000.

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
