# Peer review of "A Whole-Genome Assembly for Hyaloperonospora parasitica, A Pathogen Causing Downy Mildew in Cabbage (Brassica oleracea var. capitata L.)"

_jof, 2023, doi:10.3390/jof9080819_

Round 1

Reviewer 1 Report

I will include a marked up version with suggested changes in English

 You need to appreciate the fact that this, as are the other downy mildew pathogens, is an oomycete.  While fungi and bacteria do interact with host plants similarly as far as infection and host defense triggers, in reading the manuscript it seems strange to see direct comparisons to fungi and bacteria.  I put notes on lines 154 an 327 that address such issues.  Michelmore has done a lot of work on genomic comparisons on many plant pathogenic oomycetes, which would likely be better sources for sequence comparisons than some used.  I added a url to one of his recent reviews.  

Suggestions for improvements made on the returned file. 

Author Response

We feel great thanks for your professional review work on our article. As you are concerned, there are several problems that need to be addressed. According to your nice suggestions, we have made extensive corrections to our previous draft, the detailed corrections are listed below.

We deeply regret the errors in our manuscript and would like to express our sincere apologies. We have taken immediate action to rectify all spelling and usage errors, ensuring the manuscript is now free from any such inaccuracies.

Point 1 : Lines 141 Finally, we obtained a genome size of 37.10 Mb with an N50 of 20,542 bp and a CG percentage of 51%(Error! Reference source not found)

Response 1: Thank you for pointing out the citation error. Due to our mistake of inserting a citation for Table 2, it caused an error in the manuscript. we have removed the incorrect citation format.

Point 2: Could these mean genome is actually quite a bit larger ?  ie, they may be present in many more copies not detected via sequencing? see https://doi.org/10.1093/g3journal/jkac340

Response 2: Thank you for pointing out this problem in manuscript. We employed de novo assembly of the genome, and the sequenced and assembled genome is at the scaffold level, which may have missed small gaps. As mentioned in the recommended article you provided, it is possible that dispersed pseudogenes may not be detected. However, the overall size of the genome can still be used for subsequent analysis. This information has been added to the discussion section.

Ponint 3 : But this is an oomycete, not a bacterium or a fungus for that matter, and there is lots of information on the basis for secreted proteins among various oomycetes, including RXLR  motifs and WY domains in secreted proteins that can be host defense triggers It is true that the host-pathogen interactions are shared.  It may be best just to say pathogens in many cases.

See Fletcher and Michelmore https://doi.org/10.1146/annurev-phyto-021622-103440 and https://doi.org/10.1371/journal.ppat.1009012

Response 3: Thank you for your suggestion. We have revised the descriptions in the document and replaced the incorrect use of "fungus" and "bacterium" with the more appropriate term "pathogens". Additionally, based on the analysis of secreted proteins, we identified 65 effectors containing the RxLR motif (Supplement Table 3). Phylogenomic analysis of these RxLR effectors classified them into three clusters(Supplement Figure 3). These results have beed added to the manuscript.

Reviewer 2 Report

In the reviewed manuscript a high-quality reference genome of Hyaloperonospora parasitica, the causal agent of downy mildew in cabbage, is presented and comprehensively analyzed. A great number of pathogenesis related genes (i.e. genes involved in host-pathogen interaction, carbohydrate active enzyme-encoding genes, genes encoding membrane transport and effector proteins) was revealed through functional annotation analysis using appropriate bioinformatic tools. The significant differences among the genomes of three Brassicaceae parasitic fungi, H. parasitica, H. arabidopsidis and H. brassicae and their difference from the were demonstrated.

 Comments and recommendations:

The manuscript contains stylistic, grammatical and terminological errors which complicate the understanding the material. Please find recommendations below.

1.        Please extend the content of section 5. Conclusions. It is too brief and contains only general statements. Please indicate which conclusions can be important for the understanding mechanisms of interaction of H. parasitica and host plant can be done based on the results obtained.

2.        Please correct the latin name Brassica oleracea L. var. capitata in the title (change to Brassica oleracea var. capitate L.).

3.        The abstract contains incorrect expression: “Notably, 35 of these genes are conserved across multiple H. parasitica strain BJ2020”. What does it mean “multiple strain” in the context? In addition, the term “conserve” is not appropriate when it comes to genes commonly revealed using different databases.

4.        Lines 33-34 – the phrase “high economy” is unclear. Probably, the authors mean “high economic values (importance)”. Please edit.

5.        Line 34 – The phrase “However, cabbage could be infected by a variety of plant diseases” is incorrect. The term “infection” refers to a pathogenic agent (not to the disease).

6.        Lines 114, 118 – Please replace “repeat sequences” by “repeated sequences”.

7.        Line 151 – Please replace “lone” by “long”.

8.        Lines 169, 173 – The phrases “the strain BJ2020 is involved in various cellular processes” and “the strain BJ2020 173 is likely involved in important cellular processes” are unclear. The strain con not be involved in the processes. Please edit.

9.        Lines 240-242, 246,251, 311 – Please use “Brassicaceae” instead of “Cruciferous” (the old latin name of the family, not accepted at the present time).

10.    Please correct (extend) the names of tables 2 and 3.

11.    Please correct the legends to the Figures 1-3. It is reasonable to refer to the genome of H. parasitica strain BJ2020”.

12.    Please give a list with the legends of Supplementary figures and table name.

Author Response

Thank you for your careful reading, helpful comments, and constructive suggestions, which has significantly improved the presentation of our manuscript.

According to your suggestions, we have made modifications to the content of the manuscript, which is as follows:

  1. Please extend the content of section 5. Conclusions. It is too brief and contains only general statements. Please indicate which conclusions can be important for the understanding mechanisms of interaction of H. parasitica and host plant can be done based on the results obtained.

Response 1: We gratefully appreciate for your valuable comment. We have modified the conclusion as follows: In this study, we obtained a high-quality reference genome of H. parasitica. The whole-genome sequence of H. parasitica BJ2020 provides an important reference for subsequent transcriptomic, proteomic and metabolomic research. Besides, we identified 9991 protein-coding genes, 237 tRNAs and 13 rRNAs. GO, KEGG, and KOG annotation among these protein-coding genes indicated that most of the genes in the H. parasitica BJ2020 genome are related to cellular processes. Annotation results from the PHI database, CAZys database and other feature databases implied that reverse transcription genes may play important roles in the interaction between H. parasitica and host plants. Our analysis of H. parasitica strain BJ2020 secreted protein, PHI database and CAZys database annotations suggested that the overlapping 35 genes and 65 RxLR effectors may represent critical factors for cabbage infected by H.parasitica.This research enriched the resources of cabbage downy mildew and provided a theoretical basis for the subsequent study of H. parasitica infection and the mechanism of cabbage downy mildew disease resistance.

  1. Please correct the latin name Brassica oleracea L. var. capitata in the title (change to Brassica oleracea var. capitate L.).

Response 2: Thank you for pointing out the incorrect use of the latin name. we have changed it to Brassica oleracea var. capitate L.

3.The abstract contains incorrect expression: “Notably, 35 of these genes are conserved across multiple H. parasitica strain BJ2020”. What does it mean “multiple strain” in the context? In addition, the term “conserve” is not appropriate when it comes to genes commonly revealed using different databases.

Response 3: We feel sorry for the misunderstanding brought to the you. We have modified the statement as follows: Notably , 35 genes are annotated in all these databases, suggesting that they may be closely related to the pathogenicity of H. parasitica strain BJ2020.

4.Lines 33-34 – the phrase “high economy” is unclear. Probably, the authors mean “high economic values (importance)”. Please edit.

Response 4: Thank you for pointing out this problem in manuscript. We have modified the statement as follows: Cabbage (Brassica oleracea L. var. capitata) is an important vegetable cultivated in many countries  around the world due to its high economy and nutrition values.

  1. Line 34 – The phrase “However, cabbage could be infected by a variety of plant diseases” is incorrect. The term “infection” refers to a pathogenic agent (not to the disease).

Response 5: We are very sorry for our incorrect writing. We have modified the statement as follows: However, cabbage can be infected by a variety of pathogens, which can lead to a series of disease including clubroot, fusarium wilt, black rot and so on.

  1. Lines 114, 118 – Please replace “repeat sequences” by “repeated sequences”.

Response 6: We gratefully appreciate for point out the error. We have replaced “repeat sequences” by “repeated sequences” in Lines 144 and Line 118.

  1. Line 151 – Please replace “lone” by “long”.

Response 7: We are very sorry for our incorrect writing. We have replaced “lone” by “long” in Lines 151.

  1. Lines 169, 173 – The phrases “the strain BJ2020 is involved in various cellular processes” and “the strain BJ2020 173 is likely involved in important cellular processes” are unclear. The strain con not be involved in the processes. Please edit.

Response 8: Thank you for pointing out this problem in manuscript. We have modified it as follows: “These enriched GO terms suggested that there are lots of genes of strain BJ2020 involved in various cellular processes, such as metabolism and binding.” and “These top five enriched KEGG pathways indicate that most of the genes in strain BJ2020 are likely involved in important cellular processes, such as signal transduction, cell growth and death, and endocrine signaling, as well as fundamental metabolic pathways, such as translation and carbohydrate metabolism.”  

  1. Lines 240-242, 246,251, 311 – Please use “Brassicaceae” instead of “Cruciferous” (the old latin name of the family, not accepted at the present time).

Response 9: Considering your suggestion, we have used “Brassicaceae” instead of “Cruciferous” in manuscript.

  1. Please correct (extend) the names of tables 2 and 3.

Response 10: Thank you for your suggestion. We have extended the name of table 2 as “Table 2 Statistical results of gene functional annotation in functional databases of H. parasitica BJ2020” and table 3 as “Statistical results of comparison between the genome of H. parasitica, H. brassicae and H. arabidopsidis”

  1. Please correct the legends to the Figures 1-3. It is reasonable to refer to the genome of H. parasitica strain BJ2020”.

Response 11: Considering your suggestion, we have used modified the legends to the Figures1-3 as follows: “Figure 1 GO annotation of the genome of H. parasitica strain BJ2020”, “ Figure 2 KEGG pathway annotation of the genome of H. parasitica strain BJ2020”, “Figure 3 KOG annotation of the genome of H. parasitica strain BJ2020”

  1. Please give a list with the legends of Supplementary figures and table name.

Response 12: Thank you for your suggestion. We have listed with the legends of Supplementary figures and tables name in Supplement table 3.

We are grateful for your time and effort in reviewing our manuscript. We hope that our revised manuscript adequately addresses all concerns and meets the requirements for publication.

Reviewer 3 Report

Dear Authors,

Reviewer comments JoF-246438

The manuscript entitled „A whole-genome assembly for Hyaloperenospora parasitica, a pathogen causing downy mildew in cabbage (Brassica oleracea L. var. capitata)“ represents a useful study aimed at a sequenation, phylogenomic analysis, and genes functional characterization in the genome of downy mildew Hyaloperenospora parasitica pathogen in cabbage (Brassica oleracea var. capitata). The manuscript provides novel information on Hyaloperenospora parasitica genome as well as its comparison with related species H. parasitica, H. brassicae, and H. arabidopsidis including a phylogenomic analysis of 20 downy mildew-causing pathogens, and functional annotation of the identified genes. I can therefore recommend the present manuscript for publication in Journal of Fungi.

However, I have a few major comments and some minor formal comments on the present manuscript version which have to be addressed by the authors prior to the manuscript publication.

Major comments:

1/ Materials and methods, H. parasitica strain BJ2020 isolation: The precise location of the greenhouse in the Institute of Vegetables and Flowers, Chinese Academy of Agricultural Sciences has to be given, i.e., the name of the city where this institute is located and the H. parasitica strain used in the study was isolated has to be given.

2/ In Figure 5 providing a phylogenetic tree of 20 downy-mildew pathogens, appropriate statistics has to be added to the presented phylogenomic analysis, i.e., bootstrap values indicating probability per 1000 replicates per each branching point and appropriate scale bar have to be added to the presented phylogeentic tree.

Minor (formal) comments:

1/ Abbreviations list has to be added to the manuscript: There are relatively many abbreviations used in the manuscript which may not be familiar for the readers. Although they are explained directly in the text, I think that it would be more convenient for the readers to add a separate Abbreviations list to the manuscript.

2/ Formal comments related to terminology, English language and style:

Introduction, line 33: Correct the typing error in the term „countries“, not „counties“.

Line 142: A relevant reference has to be added following the statement „Finally, we obtained a genome size of 37.10 Mb with an N50 of 20,542 bp and a CG percentage of 51%....“

Line 166: Add a space between the words „process“ and „and“.

Line 213: Remove the words „but limited to“ and modify the statement as follows: „These proportions of different CAZys suggest that the identified enzymes may have diverse roles in the degradation and modififcation of carbohydrates including the breakdown of complex plant cell wall materials and the modification of glycans on proteins and lipids.“

Line 229: Please, check the terminology and correct it. In the statement „Among these putative secreted proteins, we identified 224 cytoplasmic effectors and 52 apoplastic effectors, the number of cytoplasmic effectors far exceeded the number of apoptotic effectors (Figure 4D).“ the words „apoptotic effectors“ should be most probably modified to „apoplastic effectors“ since there is the data on the number of cytoplasmic and apoplastic (secreted) proteins but there is no information on apoptosis, i.e., programmed cell death (PCD) given elsewhere in the manuscript.

Discussion, line 326: Correct the verb form „to affect“ instead of „to affects“ in the statement „…related to the degradation and synthesis of chitinase, cellulase, and hemicellulase to affect plant cell walls…“

Discussion, line 328: Write „a critical role“, not „an critical role“ in the statement „The effectors play a critical role in the infection of plants by pathogenic bacteria.“

Line 337: Modify the words „into the infeced mechanism“ to „the infection mechanism“ in the statement „All these results can help build a bridge for probing into the infection mechanism of H. parasitica strain BJ2020.“

Final recommendation: Reconsider after a major revision.

Dear Authors,

Reviewer comments JoF-246438

The manuscript entitled „A whole-genome assembly for Hyaloperenospora parasitica, a pathogen causing downy mildew in cabbage (Brassica oleracea L. var. capitata)“ represents a useful study aimed at a sequenation, phylogenomic analysis, and genes functional characterization in the genome of downy mildew Hyaloperenospora parasitica pathogen in cabbage (Brassica oleracea var. capitata). The manuscript provides novel information on Hyaloperenospora parasitica genome as well as its comparison with related species H. parasitica, H. brassicae, and H. arabidopsidis including a phylogenomic analysis of 20 downy mildew-causing pathogens, and functional annotation of the identified genes. I can therefore recommend the present manuscript for publication in Journal of Fungi.

However, I have a few major comments and some minor formal comments on the present manuscript version which have to be addressed by the authors prior to the manuscript publication.

Major comments:

1/ Materials and methods, H. parasitica strain BJ2020 isolation: The precise location of the greenhouse in the Institute of Vegetables and Flowers, Chinese Academy of Agricultural Sciences has to be given, i.e., the name of the city where this institute is located and the H. parasitica strain used in the study was isolated has to be given.

2/ In Figure 5 providing a phylogenetic tree of 20 downy-mildew pathogens, appropriate statistics has to be added to the presented phylogenomic analysis, i.e., bootstrap values indicating probability per 1000 replicates per each branching point and appropriate scale bar have to be added to the presented phylogeentic tree.

Minor (formal) comments:

1/ Abbreviations list has to be added to the manuscript: There are relatively many abbreviations used in the manuscript which may not be familiar for the readers. Although they are explained directly in the text, I think that it would be more convenient for the readers to add a separate Abbreviations list to the manuscript.

2/ Formal comments related to terminology, English language and style:

Introduction, line 33: Correct the typing error in the term „countries“, not „counties“.

Line 142: A relevant reference has to be added following the statement „Finally, we obtained a genome size of 37.10 Mb with an N50 of 20,542 bp and a CG percentage of 51%....“

Line 166: Add a space between the words „process“ and „and“.

Line 213: Remove the words „but limited to“ and modify the statement as follows: „These proportions of different CAZys suggest that the identified enzymes may have diverse roles in the degradation and modififcation of carbohydrates including the breakdown of complex plant cell wall materials and the modification of glycans on proteins and lipids.“

Line 229: Please, check the terminology and correct it. In the statement „Among these putative secreted proteins, we identified 224 cytoplasmic effectors and 52 apoplastic effectors, the number of cytoplasmic effectors far exceeded the number of apoptotic effectors (Figure 4D).“ the words „apoptotic effectors“ should be most probably modified to „apoplastic effectors“ since there is the data on the number of cytoplasmic and apoplastic (secreted) proteins but there is no information on apoptosis, i.e., programmed cell death (PCD) given elsewhere in the manuscript.

Discussion, line 326: Correct the verb form „to affect“ instead of „to affects“ in the statement „…related to the degradation and synthesis of chitinase, cellulase, and hemicellulase to affect plant cell walls…“

Discussion, line 328: Write „a critical role“, not „an critical role“ in the statement „The effectors play a critical role in the infection of plants by pathogenic bacteria.“

Line 337: Modify the words „into the infeced mechanism“ to „the infection mechanism“ in the statement „All these results can help build a bridge for probing into the infection mechanism of H. parasitica strain BJ2020.“

Final recommendation: Reconsider after a major revision.

Author Response

We would like to thank you for your careful reading, helpful comments, and constructive suggestions, which has significantly improved the presentation of our manuscript.

We have carefully considered all comments from the yours and revised our manuscript accordingly. The manuscript has also been double-checked, and the typos and grammar errors we found have been corrected. In the following section, we summarize our responses to each comment from the reviewers. We believe that our responses have well addressed all concerns from the reviewers. We hope our revised manuscript can be accepted for publication.

Major comments 1:/: Materials and methods, H. parasitica strain BJ2020 isolation: The precise location of the greenhouse in the Institute of Vegetables and Flowers, Chinese Academy of Agricultural Sciences has to be given, i.e., the name of the city where this institute is located and the H. parasitica strain used in the study was isolated has to be given.

Response 1.1/: Thank you for your suggestion. We have modified it as follows: H. parasitica strain BJ2020 was isolated from the cabbage inbred line “2020-w5”, cultivated in the greenhouse of the Institute of Vegetables and Flowers, Chinese Academy of Agricultural Sciences, Beijing, China. Fresh downy mildew conidia were isolated from naturally infected leaves in the field using a method as previously described. Subsequently conidial suspension were sprayed on the seedlings of "2020-w5". After grown in a greenhouse under a 16 h light/8 h dark cycle for 6 d then placed under high humidity in the dark for 24 h, the newly formed sporangia will germinate. The inoculated plants showed heavy necrosis with sporulation dispersed over the entire leaf surface. The above procedure was repeated over several times, finally the purified H. parasitica isolate was obtained, which were named BJ2020.

Major comments 2: In Figure 5 providing a phylogenetic tree of 20 downy-mildew pathogens, appropriate statistics has to be added to the presented phylogenomic analysis, i.e., bootstrap values indicating probability per 1000 replicates per each branching point and appropriate scale bar have to be added to the presented phylogeentic tree.

Response 1.2/: According to your suggestion, bootstrap values and a scale bar have been added to the phylogenetic tree image.

Minor (formal) comments

1/:Abbreviations list has to be added to the manuscript: There are relatively many abbreviations used in the manuscript which may not be familiar for the readers. Although they are explained directly in the text, I think that it would be more convenient for the readers to add a separate Abbreviations list to the manuscript.

Response 2.1/: Thank you for your suggestion. We have added separate Abbreviations list as supplement table2.

Minor (formal) comments 2/: Formal comments related to terminology, English language and style:

Introduction, line 33: Correct the typing error in the term „countries“, not „counties“.

Response 2.2/: Thank you for pointing out the spelling mistake. We have corrected this error in Line 33.

Line 142: A relevant reference has to be added following the statement „Finally, we obtained a genome size of 37.10 Mb with an N50 of 20,542 bp and a CG percentage of 51%....“

Response 2.3/: Thank you for pointing out the citation error. Due to our mistake of inserting a citation for Table 2, it caused an error in the manuscript. we have removed the incorrect citation format.

Line 166: Add a space between the words „process“ and „and“.

Response 2.4/: Thank you for pointing out the error. We have made the necessary correction in line 166 of the manuscript.

Line 213: Remove the words „but limited to“ and modify the statement as follows: „These proportions of different CAZys suggest that the identified enzymes may have diverse roles in the degradation and modififcation of carbohydrates including the breakdown of complex plant cell wall materials and the modification of glycans on proteins and lipids.“

Response 2.5/: Thank you for your suggestion. We have modified the statement as follows: These proportions of different CAZys suggest that the identified enzymes may have diverse roles in the degradation and modification of carbohydrates, including the breakdown of complex plant cell wall materials and the modification of glycans on proteins and lipids.

Line 229: Please, check the terminology and correct it. In the statement „Among these putative secreted proteins, we identified 224 cytoplasmic effectors and 52 apoplastic effectors, the number of cytoplasmic effectors far exceeded the number of apoptotic effectors (Figure 4D).“ the words „apoptotic effectors“ should be most probably modified to „apoplastic effectors“ since there is the data on the number of cytoplasmic and apoplastic (secreted) proteins but there is no information on apoptosis, i.e., programmed cell death (PCD) given elsewhere in the manuscript.

Response 2.6/: Thank you for pointing out the terminology error. We have modified the statement as follows: Among these putative secreted proteins, we identified 224 cytoplasmic effectors and 52 apoplastic effectors, the number of cytoplasmic effectors far exceeded the number of apoplastic effectors(Figure 4D).

Discussion, line 326: Correct the verb form „to affect“ instead of „to affects“ in the statement „…related to the degradation and synthesis of chitinase, cellulase, and hemicellulase to affect plant cell walls…“

Response 2.7/: Thank you for pointing out the word error. We have modified the statement as follows: It is reported that GH and GT class gene were mainly related to the degradation and synthesis of chitinase, cellulase, and hemicellulase to affect plant cell walls.

Discussion, line 328: Write „a critical role“, not „an critical role“ in the statement „The effectors play a critical role in the infection of plants by pathogenic bacteria.“

Response 2.8/: Thank you for pointing out the incorrect use of articles. We have modified the statement as follows: The effectors play a critical role in the infection of plants by pathogenic pathogens.

Line 337: Modify the words „into the infeced mechanism“ to „the infection mechanism“ in the statement „All these results can help build a bridge for probing into the infection mechanism of H. parasitica strain BJ2020.“

Response 2.9/: Thank you for pointing out the incorrect use of words. We have modified the statement as follows: All these results can help build a bridge for probing into the infection mechanism of H. parasitica strain BJ2020.

We are grateful for your time and effort in reviewing our manuscript. We hope that our revised manuscript adequately addresses all concerns and meets the requirements for publication.

Round 2

Reviewer 3 Report

Dear Authors,

Reviewer comments JoF-2443434.R1

The revised manuscript entitled „A whole-genome assembly for Hyaloperenospora parasitica, a pathogen causing downy mildew in cabbage (Brassica oleracea var. Capitata L.)“ was significantly improved by the authors in accordance with my previous comments. However, I still have some minor comments on the revised manuscript which are given below:

1/ In Materials and methods, web addresses and dates of download have to be added for all databases listed in part 2.4 Gene prediction and functional annotation.

2/ Formal comments related to English language and style:

Line 76: Add a comma following the word „Subsequently“ in the statement „Subsequently, conidial suspension…“

Line 80: Add a comma following the word „finally“ in the statement „…finally, the purified H. parasitica isolate was obtained which was named BJ2020.“

Line 335: Use just the term „pathogens“ instead of „pathogenic pathogens“ in the statement „Cytoplasmic effectors may act as a target or transfer of pathogens…“

Final recommendation: Accept after a minor revision.

Dear Authors,

Reviewer comments JoF-2443434.R1

The revised manuscript entitled „A whole-genome assembly for Hyaloperenospora parasitica, a pathogen causing downy mildew in cabbage (Brassica oleracea var. Capitata L.)“ was significantly improved by the authors in accordance with my previous comments. However, I still have some minor comments on the revised manuscript which are given below:

1/ In Materials and methods, web addresses and dates of download have to be added for all databases listed in part 2.4 Gene prediction and functional annotation.

2/ Formal comments related to English language and style:

Line 76: Add a comma following the word „Subsequently“ in the statement „Subsequently, conidial suspension…“

Line 80: Add a comma following the word „finally“ in the statement „…finally, the purified H. parasitica isolate was obtained which was named BJ2020.“

Line 335: Use just the term „pathogens“ instead of „pathogenic pathogens“ in the statement „Cytoplasmic effectors may act as a target or transfer of pathogens…“

Final recommendation: Accept after a minor revision.

Author Response

We feel great thanks for your professional review work on our article. As you are concerned, there are several problems that need to be addressed. According to your nice suggestions, we have made extensive corrections to our previous draft, the detailed corrections are listed below.

We deeply regret the errors in our manuscript and would like to express our sincere apologies. We have taken immediate action to rectify all spelling and usage errors, ensuring the manuscript is now free from any such inaccuracies.

Point 1/: In Materials and methods, web addresses and dates of download have to be added for all databases listed in part 2.4 Gene prediction and functional annotation.

Response 1: Thank you for your suggestion. We have added web addresses for all databases listed in part 2.4 Gene prediction and functional annotation.

Ponit 2/:  Formal comments related to English language and style:

Line 76: Add a comma following the word „Subsequently“ in the statement „Subsequently, conidial suspension…“

Line 80: Add a comma following the word „finally“ in the statement „…finally, the purified H. parasitica isolate was obtained which was named BJ2020.“

Line 335: Use just the term „pathogens“ instead of „pathogenic pathogens“ in the statement „Cytoplasmic effectors may act as a target or transfer of pathogens…“

Reaponse 2: We are very sorry for our incorrect writing. We have added a comma following the word “Subsequently” in the statement “Subsequently, conidial suspension…” and a comma following the word “finally” in the statement “…finally, the purified H. parasitica isolate was obtained which was named BJ2020.” Besides, we have deleted the word “pathogenic” in the statement “Cytoplasmic effectors may act as a target or transfer of pathogenic pathogens…“

We are grateful for your time and effort in reviewing our manuscript. We hope that our revised manuscript adequately addresses all concerns and meets the requirements for publication.
